# Nano-Scale Pore Structure and Its Multi-Fractal Characteristics of Tight Sandstone by $N_2$ Adsorption/Desorption Analyses: A Case Study of Shihezi Formation from the Sulige Gas Filed, Ordos Basin, China

**Zhelin Wang**[ID]**, Xuewei Jiang, Mao Pan and Yongmin Shi \***

School of Earth and Space Sciences, Peking University, Beijing 100871, China; zhelin_wang@pku.edu.cn (Z.W.); jiangxw@pku.edu.cn (X.J.); panmao@pku.edu.cn (M.P.)

**\*** Correspondence: sym@pku.edu.cn

**Abstract:** Fractal dimension is a critical parameter to evaluate the heterogeneity of complex pore structure in tight sandstone gas and other low permeability reservoirs. To quantify the fractal dimension of tight sandstone at various pore size classes and evaluate their implications on mineral composition and nano pore structure parameters, we conducted an integrated approach of $N_2$ adsorption/desorption experiment ($N_2$-GA), X-ray diffraction (X-RD), and field emission scanning electron microscopy (FE-SEM) on Sulige tight sandstone reservoirs. By comparing the nine types of fractal dimensions calculated from $N_2$ adsorption data, we put forward the concept of "concentrated" fractal dimensions and "scattered" fractal dimensions ($D_{N2}$, $D_{N3}$, $D_{N5}$, $D_{N7}$ and $D_{N8}$) for the first time according to its concentration extent of distribute in different samples. Result shows that mineral composition has a significant influence of a different level on specific surface area (SSA), pore volume (PV), and fractal dimensions ($D_N$), respectively, where the "scattered" fractal dimension is more sensitive to certain specific property of the reservoir, including mineral content and the specific surface area contribution rate ($S_r$) of type II mesopores (Mesopore-II: 10~50nm). In addition, three type of hysteresis loops were distinguished corresponding to different pore shape combination of $N_2$-GA isotherm curve, which reveals that pore structure heterogeneity is mainly controlled by inkbottle-shaped pores and the volume contribution rate ($V_r$) of mesopores in this study area. These findings could contribute to a better understanding of the controlling effect of pore heterogeneity on natural gas storage and adsorption.

**Keywords:** tight sandstone; fractal dimension; nanopore structure; Sulige gas field; Shihezi Formation

## 1. Introduction

Energy sources around the world have been changing from solid (firewood and coals) and liquid (petroleum) states to gas state (natural gas) [1], where tight sandstone reservoirs have been playing an important role and exploited industrially in many countries as an effective storage of natural gas, which is a complicated porous material with a highly heterogeneous nanopore structure [2–8]. Owing to the complexity of the its pore-throat system, pore structure and its fractal characteristics analysis become the core work of reservoir quality evaluation, which includes the research content of specific surface area (SSA), pore volume (PV), pore size distribution (PSD), pore morphology, and anisotropy [9–11].

Due to the complicated 3D characteristics of a tight sandstone pore-throat system, it is difficult for traditional Euclidean geometry to describe the pore morphological characteristics precisely. Fractal

theory is an effective approach to evaluating the surfaces roughness and complexity of irregular pores, which was first proposed by Benoit B. Mandelbrot in 1983 [12–17], where fractal dimension is an essential feature parameter. It can quantitatively characterize nanoscale pore heterogeneity and make a better understanding of the pore microstructures complexity comprehensively, with a value usually distributed between two and three. In general, fractal dimension value often reflects the heterogeneity of the pore-throat system including connectivity, coordination, and the complexity of pore shape, where a higher value indicates a more complex pore-throat system [18,19]. In addition, fractal theory also reveals that the rock surface displays a property of "self-similar regularity" at molecular scales and exhibits a similar structure regularity over 3-4 orders of magnitude resolution scales in length between 1 nm and 100 μm [20,21].

In recent years, with the development of pore structure characterization technology, there are four established methods to calculate fractal dimensions based on experimental data: fluid injection technique, gas adsorption [22,23], image analysis [18,19], and no-destructive methods through CT scanning [24,25], where fluid injection technique includes low-field nuclear magnetic resonance (NMR) [21,26–28] and mercury intrusion porosimetry (MIP) [29–31]. In terms of $N_2$ absorption/desorption data, numerous studies have applied Frenkel–Halsey–Hill (FHH) model to study the fractal characteristics of nanopore structure in different reservoir, including coal-bearing organic shale in different facies [20,21,32–38], coal [23,39–42], or tight sandstone [29,30], which is proven to be an efficient method to capture fractal dimensions aim at nanoscale pores (2–400 nm) [43]. Previous research usually obtained fractal dimensions by the unsegmented method, or capture two fractal dimensions D1 ($P/P_0$: 0–0.5) and D2 ($P/P_0$: 0.5–1) represents the fractals generated by volume irregularity and pore structures, respectively [18]. Moreover, many of them have already made a comparative study between $N_2$ adsorption with NMR or MIP [30,37,44], whereas the scientificity and validity of different fractal dimensions obtained from nitrogen adsorption data have not been distinguished, and its influence factors are still not explicit.

Therefore, in this paper, $N_2$ adsorption/desorption experiment ($N_2$-GA) was utilized to accurately characterize the nanoscale pore heterogeneity, morphology and structure parameters of 11 different ranks of tight sandstone from the Sulige gas field. Fractal characteristics of pore throats with different sizes were specified according to the FHH model. Furthermore, field emission scanning electron microscopy (FE-SEM) was used to help make more explicit the pore morphology features, petrographic characteristic and the relationship between them. In addition, we used X-ray diffraction (X-RD) to characterize the mineral composition of all samples accurately. First, pore morphological characteristics and mineral composition were identified by FE-SEM and X-RD. Then, based on $N_2$-GA data of all 11 samples, we effectively captured the PSD, SSA, and PV, and nine types of fractal dimensions were calculated to understand the complexity of pore microstructures comprehensively, which can clarify the effect of mineral composition and pore structure on different types of fractal dimensions, respectively. Hence, the major aims of this study were to (1) quantitatively characterize the nanoscale pore heterogeneity of tight sandstone in He 8 Member at different size classes and (2) clarify the effect factor of different types of fractal dimensions, respectively.

## 2. Samples and Methods

### 2.1. Samples and Geological Setting

The tight sandstone gas field in the Sulige area is located in the northwest of Yishan slope in the Ordos Basin, which is a typical example of tight gas in China. The Ordos Basin is the second largest sedimentary basin in China (Figure 1a), which is located in the western part of North China Craton, and is a large multi-cycle superimposed basin with overall uplift and simple geotectonics [45,46]. According to previous studies, the eighth member of the Middle Permian Lower Shihezi Formation (He 8 Member) is generally considered as a set of deposits of braided river-shallow delta facies, with a

larger variation range of hydrodynamic conditions. Under a wide and gentle geotectonic background, the channel changes and overlaps in various periods, forming a complex pattern structure [47].

Based on core observation, 11 tight sandstone samples with different physical characteristics were collected from the He 8 Member in the Ordos Basin, which is a main gas production layer of the Sulige large gas province (Figure 1b). In order to determine the heterogeneity of nanoscale pore structure, we collected core samples from one appraisal well where the burial depth varies from 3326 m to 3358 m. Figure 2 shows the generalized stratigraphic columns of the Shihezi Formation and sampling depth and its corresponding logging profile and gamma (GR) curve.

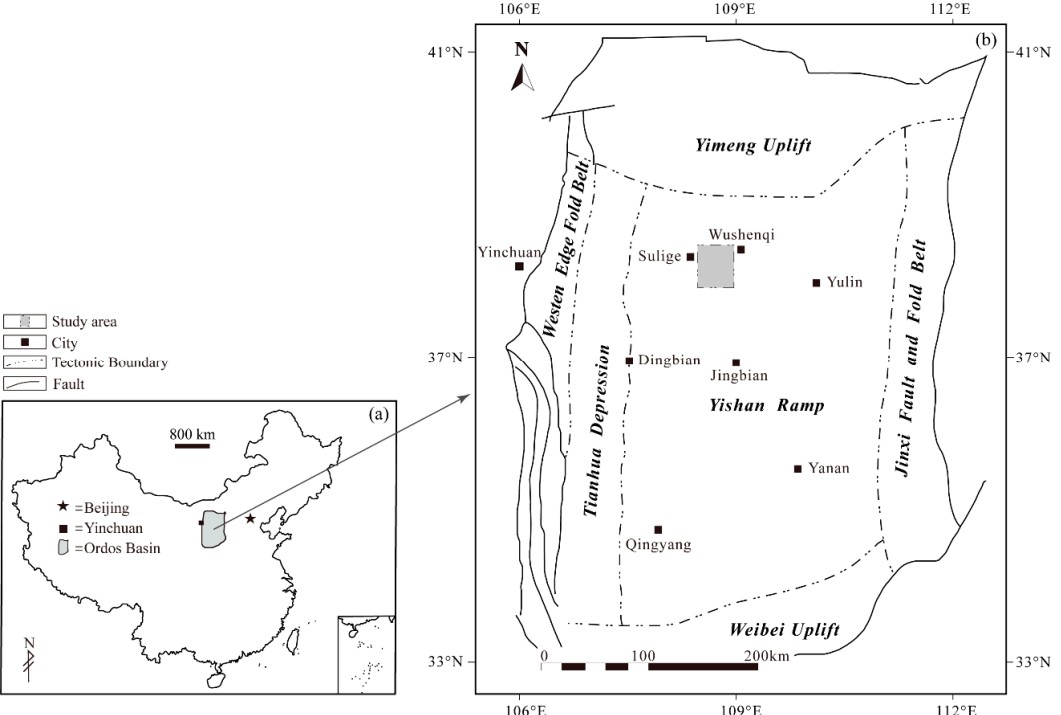

**Figure 1.** Location of the study area showing in the tectonic map of Ordos Basin. (**a**) Geographic location of the Ordos Basin; (**b**) geographic location of the study area.

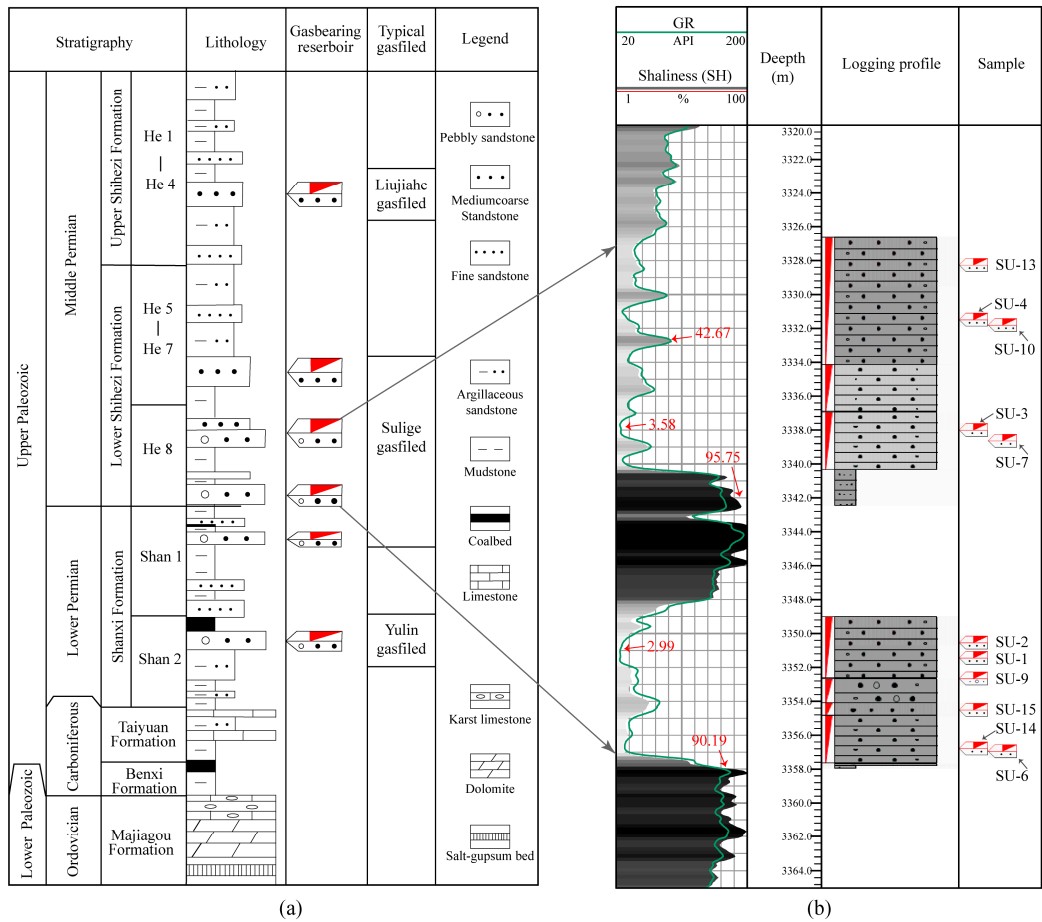

**Figure 2.** Schematic diagram of sampling location. (**a**) Generalized stratigraphic columns of the Shihezi Formation (modified from reference [47]); (**b**) sampling depth and its corresponding logging profile and gamma (GR) curve.

## 2.2. Experimental Methods

All 11 samples were analyzed using X-RD analysis, FE-SEM observation, and $N_2$-GA. First, pore morphological characteristics and mineral composition were identified by XRD and FE-SEM. Then built on $N_2$ adsorption data of all 11 samples, whicheffectively captured PSD, SSA, and PV. Three types hysteresis loop were distinguished, and nine types of fractal dimensions were calculated to understand the complexity of pore microstructures comprehensively.

$N_2$ adsorption/desorption isotherms were obtained using a Quantachrome QUADRAS-ORB SI Surface Area and Pore Size Analyzer (Quantachrome, Boynton Beach, FL, USA) –196.15 °C, with a pore size test range of 0.35–400 nm. Experimental process was conducted with a mixture of block sample and 60–80 mesh powder samples for the purpose of eliminating the experimental error caused by sample size in accordance with the Chinese government standard GB/T 19587-2017.

X-RD was employed as a supporting test for analyzing mineral composition, using powder samples that were crushed to less than 200 mesh size. The X-ray diffractometer was operated at 25 °C and 10% humidity with a step of $0.01°(2\theta)$ performed using a PANalytical X'Pert3 Powder (PANalytical, Almelo, The Netherlands), with the procedure adhering to the Chinese Oil and Gas Industry Standard SY/T5163 (2018).

The FE-SEM analysis was completed using a FEI Quanta 650 high performance multipurpose field emission-scanning electron microscopy (FEI, Hillsboro, OR, USA). Prior to observation, tight sandstone samples were polishing by an Ar-ion milling system, and the samples were then sputter-coated with chromium for conductivity. The specifications for the high-resolution imaging and analysis of

the conductive samples were acceleration voltage: 200 V–30 kV; resolution: 1.4 nm at 30 kV under environmental vacuum conditions (ESEM); and magnification: 80–6 × 1,000,000.

### 2.3. Fractal Model of N$_2$ Adsorption

The FHH model has proven to be an effective model and has been widely used to obtain the fractal dimension based on N$_2$ adsorption data [22,48–50], which can be expressed as the following double logarithm form:

$$ln(V) = k \times ln[ln(P_0/P)] + C \tag{1}$$

where $V$ is the volume of N$_2$ adsorption capacity at equilibrium pressure in cm$^3$/g, $P_0$ is the gas saturation pressure in MPa, $P$ is the gas equilibrium pressure in MPa, C is constant, and k is the slope of ln($V$) *vs.* ln($P_0$/P) linear fitting plots. On this foundation, fractal dimension ($D_N$) based on N$_2$ adsorption/desorption can be calculated by following equation:

$$D_N = k + 3 \tag{2}$$

## 3. Results

### 3.1. Minerals Composition

According to the X-RD result of investigated core samples, the mineral composition of the investigated samples shown in Table 1 and Figure 3, it can be seen that He 8 Member tight sandstone mainly consists of quartz, clay, calcite, and feldspar. Quartz abundance between 49.6% and 93.6% averages at 81.5%. Total clay content ranges from 3.0% to 31.3% with an average value of 14.0%, in which the content of chlorite is highest (from 19.0% to 77.0%, averages at 43.5%), followed by kaolinite (from 11.0% to 57.0%, averages at 28.5%), illite/smectite (from 3.0% to 37.0%, averages at 16.8%), and illite (from 6.0% to 27.0%, averages at 13.8%). These clustered intercrystalline micropores of clay minerals exist as a local enrichment form with poor connectivity, developed inside original intergranular pores wherefore the percolation capacity of the reservoir was impaired to a certain extent as well.

**Table 1.** Mineral composition based on X-ray diffraction (X-RD) analysis of the eighth member of the Middle Permian Lower Shihezi Formation (He 8 Member) tight sandstone.

| Sample ID | Mineral Composition (%) | | | | Clay Composition (%) | | | |
|---|---|---|---|---|---|---|---|---|
| | Q | F | C | Clay | I/S | I | K | C |
| SU-1 | 90.6 | 0.2 | 4.2 | 5 | 5 | 7 | 57 | 31 |
| SU-2 | 93.6 | 0.2 | | 6.2 | 15 | 12 | 11 | 62 |
| SU-3 | 93.3 | 0.2 | 0.4 | 6.1 | 21 | 15 | 38 | 26 |
| SU-4 | 89.5 | 0.2 | 3.8 | 6.5 | 3 | 13 | 48 | 36 |
| SU-6 | 87.8 | 0.4 | 1 | 10.8 | 10 | 13 | 22 | 55 |
| SU-7 | 88.2 | 0.2 | | 11.6 | | 6 | 17 | 77 |
| SU-9 | 49.6 | 0.4 | 33.5 | 16.5 | 28 | 16 | 23 | 33 |
| SU-10 | 67.2 | 1.5 | | 31.3 | 37 | 27 | 17 | 19 |
| SU-13 | 73.6 | 0.2 | 1.1 | 25.1 | 11 | 16 | 35 | 38 |
| SU-14 | 84.7 | 0.5 | 0.6 | 14.2 | 14 | 14 | 27 | 45 |
| SU-15 | 78.8 | 0.6 | | 20.6 | 24 | 13 | 18 | 45 |
| Average | 81.5 | 0.4 | 6.4 | 14.0 | 16.8 | 13.8 | 28.5 | 42.5 |

Note: Q—quartz; F—feldspar; Ca—calcite; I/S—Illite/Smectite; I—Illite; K—kaolinite; C—chlorite.

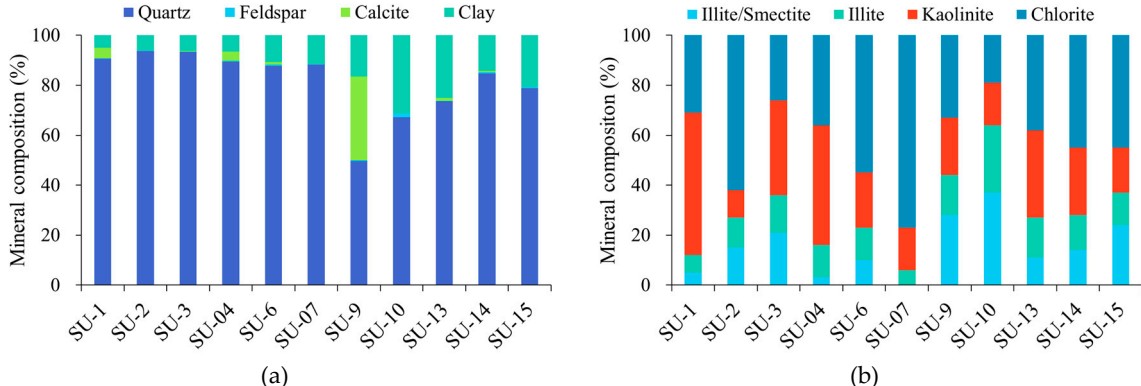

**Figure 3.** Mineral composition bar chart of He 8 Member tight sandstone. (**a**) Total rock mineral composition bar charts; (**b**) clay mineral composition bar charts.

### 3.2. Nanoscale Pore Geometry from $N_2$ Adsorption/Desorption Isotherms

The $N_2$ adsorption isotherms can be classified into six types according to the International Union of Pure and Applied Chemistry (IUPAC) (Figure 4a). Investigated tight sandstone samples belong to type IV with a hysteresis loop, which are anti "s" as a whole, and samples in this study area are mesoporous solid, where the hysteresis loop patterns are indicative of mixtures of mesopores and micropores in the samples. Furthermore, type IV $N_2$ adsorption/desorption hysteresis loops can be divided into four main types, which correspond to cylindrical pores, inkbottle-shaped pores, parallel plate shaped pores, and wedge-/slit-shaped pores (Figure 4c) [18,40,51]. Based on the pattern of hysteresis loops, the 11 He 8 Member tight sandstone samples mainly exhibit three types—H1, H2, and H3.

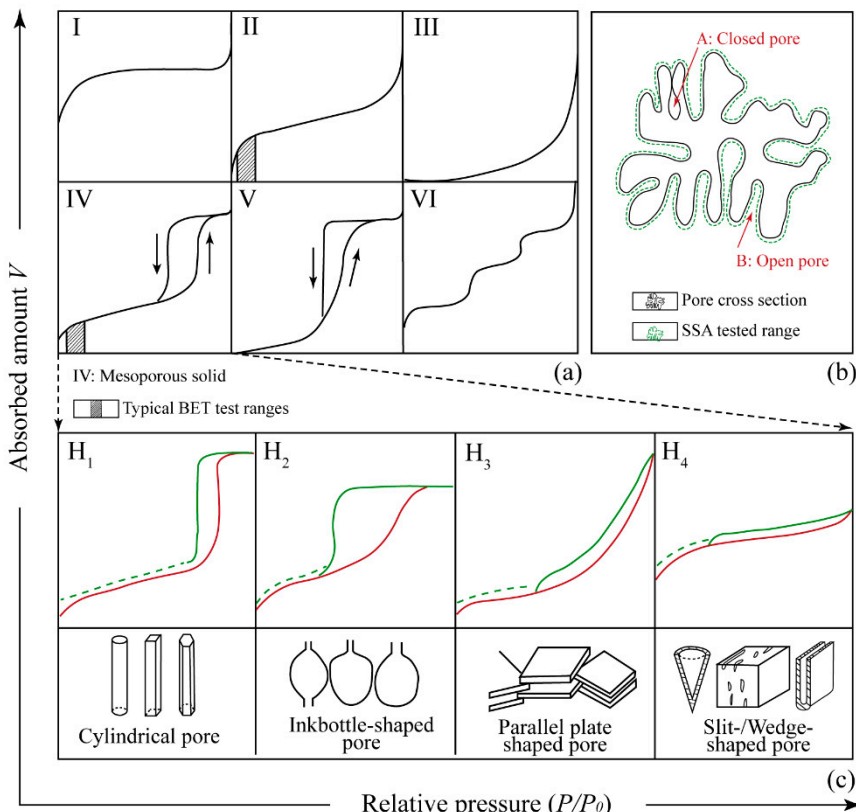

**Figure 4.** Type of adsorption isotherms and its corresponding pore types: (**a**) Type of adsorption isotherms; (**b**) specific surface area (SSA) tested range by Brunauer-Emmett-Teller (BET) test; (**c**) hysteresis loops and its corresponding pore shapes.

As the result shown in Figure 5 and Table 2, type H1 including SU-6, SU-7, and SU-14 always has a smaller amount $N_2$ adsorption volume with narrow hysteresis loops, where their adsorption curves almost completely coincide with the desorption curves and increase rapidly and synchronously at a $P/P_0 = 0.9$ approaching to 1.0. This type of loop usually corresponds to the combination of cylindrical pores and parallel plate shaped pores [40], the cylindrical pores often with the pore size distributed in 50–200 nm (macropores) and the parallel plate shaped pores usually with the pore size varies from 5–50 nm.

Analogously, the inflection point for the rapid increase of type H3 (SU-1, SU-2, SU-3, SU-4) appears around $P/P_0 = 0.8$ with a slightly wider hysteresis loop, which is the main difference from H1 type loops. This type of loop is usually related to aggregates of slit-/wedge-shaped particles giving rise to parallel plate shaped pore with the pore size varies from 2–50 nm (slit-/wedge-shaped) and 50–100 nm (parallel plate shaped pore) [51].

For type H2 represented by SU-9, SU-10, SU-13, SU-15, the inflection point visible in the desorption curves appears around $P/P_0 = 0.5$, which developed a wider hysteresis; this indicates the morphology of mesopore-I present a fine bottleneck with one side almost closed or double side open [18]. In addition, parallel plate–shaped pores and cylindrical pores usually appear in combination among these samples.

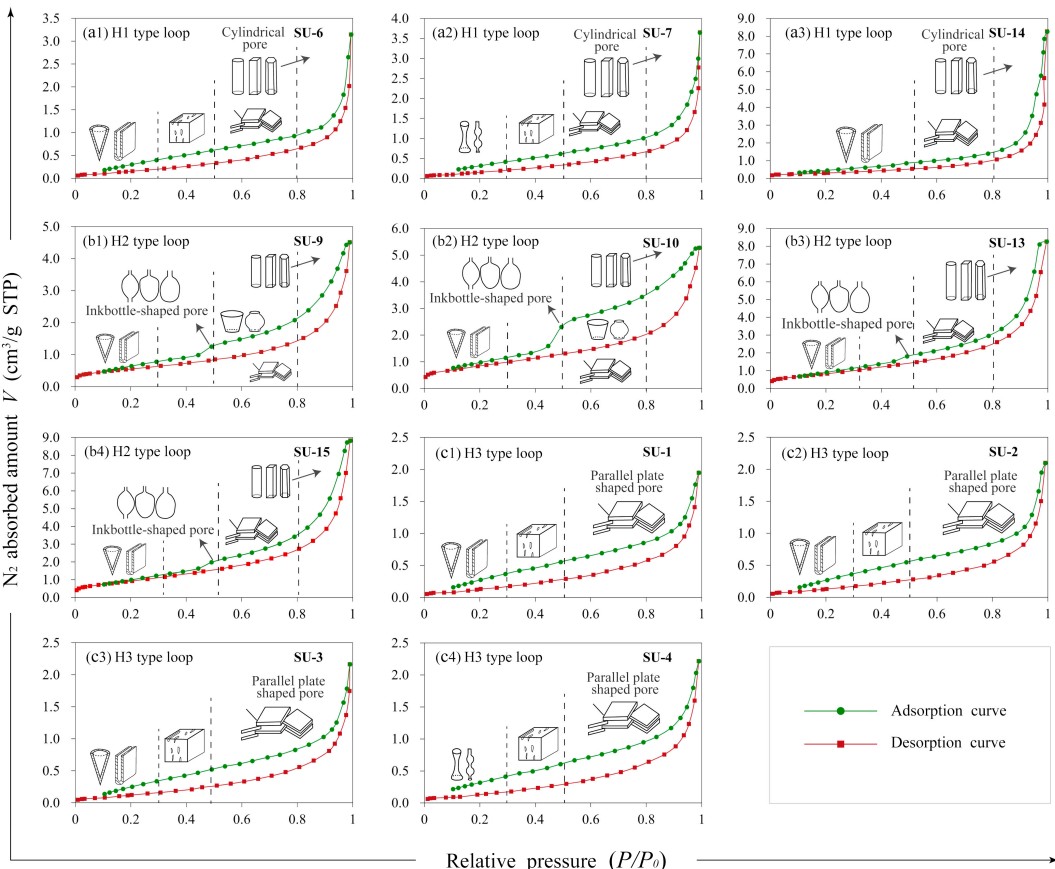

**Figure 5.** $N_2$ adsorption/desorption isotherms and the corresponding pore shapes of He 8 Member tight sandstone samples. (**a1**)–(**a3**) Samples with H1 type hysteresis loop; (**b1**)-(**b4**) samples with H2 type hysteresis loop; (**c1**)–(**c4**) samples with H3 type hysteresis loop.

### 3.3. Specific Surface Area, Pore Volume, and Pore Size Distribution

The nanopore structure parameter in this study was calculated by the BET model (SSA) [52], BJH model (PV of mesopores) [53], and HK-SF model (PV of micropores) based on $N_2$ adsorption data. It is noteworthy that the closed pores are not included in the BET test result (Figure 4b). The nanoscale pore quantitative analysis consequence of He 8 Member tight sandstone by $N_2$ adsorption-desorption

experiments can be seen in Table 2. The BET SSA ($S_{BET}$) ranges from 0.530–3.162 m$^2$/g (averages at 1.297 m$^2$/g). The BJH adsorption cumulative SSA ($S_{BJH}$) ranges from 0.925–3.858 m$^2$/g (averages at 1.641 m$^2$/g), where the average value is higher than $S_{BET}$. The BJH adsorption cumulative volume ($V_{BJH}$) ranges from 3.291–13.379 × 10$^{-3}$ cm$^3$/g (averages at 6.703 × 10$^{-3}$ cm$^3$/g) showed a good positive relationship with $S_{BET}$ ($R^2 = 0.569$) (Figure 6a) and $S_{BJH}$ ($R^2 = 0.657$) (Figure 6b). The SF micropore volume ($V_{SF}$) ranges from 1.198 ~ 0.220 ×1 0$^{-3}$cm$^3$/g with an average of 0.518 × 10$^{-3}$ cm$^3$/g, which shows a better positive relationship with $S_{BET}$ ($R^2 = 0.988$) (Figure 6c) and $S_{BJH}$ ($R^2 = 0.903$) (Figure 6d), this result indicated that micropores has a higher contribution to SSA than mesopores and macropores.

The N$_2$ maximum absorbed amount ($A_M$) of 11 sample ranges from 1.950–8.266 cm$^3$/g (averages at 4.155 cm$^3$/g), which showing an excellent positive relationship with PV ($V_{BJH}$: $R^2 = 0.997$, $V_{SF}$: $R^2 = 0.615$) (Figure 6e,f). This result revealed that N$_2$ maximum absorbed amount can represent the total PV of different samples, where the mesopores have a higher contribution. In consequence, we divided samples into three types depending on the maximum adsorption capacity, which can reflect the reservoir quality. Class A samples including SU-13, SU-14, SU-15, $A_M$ are within the range of 7.0–10.0 m$^2$/g, with an average value of 8.453 m$^2$/g among three samples. Class B samples including SU-6, SU-7, SU-9 SU-10, $A_M$ are within the range of 3.0~7.0 m$^2$/g, with an average value of 4.148 m$^2$/g. among four samples. Class C samples including SU-1, SU-2, SU-3, SU-4, $A_M$ are within the range of 0.1–3.0 m$^2$/g, with an average value of 4.148 m$^2$/g among four samples.

Among all three types samples, Class C samples only have a single form hysteresis loop type (H3 type loop). Whereas Class B developed H1 type loop (SU-6, SU-7) and H2 type loop (SU-9, SU-10) simultaneously. This phenomenon is related to the complex morphological features of mesopores in samples with H2 type loop as well as a smaller volume of the micropore ($V_{SF}$) in samples with H1 type loop. Furthermore, as listed in Table 2, the total adsorption average pore width ($W_T$) ranges from 10.329–45.9839 nm with an average of 24.036 nm, where samples with H1 type loop often have a larger total pore average width, while samples with H2 type loop always have a smaller one. Based on the corresponding pore shapes to H2 type loops, revealed a large amount of bowl-shaped pores in mesopores, which is the transitional form from an inkbottle-shaped pores to parallel plate shaped pores, such as SU-9 (Figure 5b1) and SU-10 (Figure 5b2).

**Table 2.** Results of pores surface area, volume and pore width by N$_2$ adsorption/desorption experiment of He 8 Member tight sandstone.

| Sample ID | $A_M$ (cm$^3$/g) | Specific Surface Area (m$^2$/g) | | Pore Volume (×10$^{-3}$ cm$^3$/g) | | Pore Width (nm) | | | | Classfication | Hysteresis Loop Type |
|---|---|---|---|---|---|---|---|---|---|---|---|
| | | $S_{BET}$ | $S_{BJH}$ | $V_{BJH}$ | $V_{SF}$ | $W_T$ | $W_{BJH\_ave}$ | $W_{BJH\_med}$ | $W_{SF\_med}$ | | |
| SU-1 | 1.950 | 0.597 | 1.014 | 3.291 | 0.250 | 20.212 | 12.983 | 2.040 | 1.835 | C | H3 |
| SU-2 | 2.100 | 0.548 | 0.944 | 3.503 | 0.247 | 23.726 | 14.840 | 2.035 | 1.967 | C | H3 |
| SU-3 | 2.166 | 0.530 | 0.925 | 3.593 | 0.220 | 25.271 | 15.547 | 3.509 | 1.728 | C | H3 |
| SU-4 | 2.217 | 0.594 | 1.044 | 3.725 | 0.334 | 23.104 | 14.271 | 2.801 | 1.718 | C | H3 |
| SU-6 | 3.148 | 0.697 | 1.124 | 5.167 | 0.288 | 27.931 | 18.379 | 2.814 | 1.727 | B | H1 |
| SU-7 | 3.652 | 0.683 | 1.188 | 5.987 | 0.326 | 33.094 | 20.151 | 2.033 | 1.728 | B | H1 |
| SU-9 | 4.512 | 2.012 | 1.932 | 7.043 | 0.746 | 13.877 | 14.583 | 3.976 | 1.108 | B | H2 |
| SU-10 | 5.279 | 3.162 | 2.862 | 8.147 | 1.198 | 10.329 | 11.387 | 2.099 | 1.067 | B | H2 |
| SU-13 | 8.261 | 3.038 | 3.858 | 13.379 | 1.055 | 16.823 | 13.872 | 2.819 | 1.071 | A | H2 |
| SU-14 | 8.266 | 1.112 | 1.519 | 13.198 | 0.518 | 45.989 | 34.762 | 2.062 | 1.728 | A | H1 |
| SU-15 | 8.831 | 3.772 | 4.368 | 13.404 | 1.272 | 13.686 | 12.275 | 2.498 | 1.104 | A | H2 |

Note: $A_M$–N$_2$ Maximal absorbed amount; $S_{BET}$—BET SSA; $S_{BJH}$—BJH Adsorption cumulative SSA; $V_{BJH}$—BJH Adsorption cumulative PV; $V_{SF}$—SF Micropore PV; $W_T$—Total adsorption average pore width; $W_{BJH\_ave}$—BJH Adsorption average pore width; $W_{BJH\_med}$—BJH Adsorption median pore width; $W_{SF\_med}$—SF Median pore width; A—Class A; B—Class B; C—Class C.

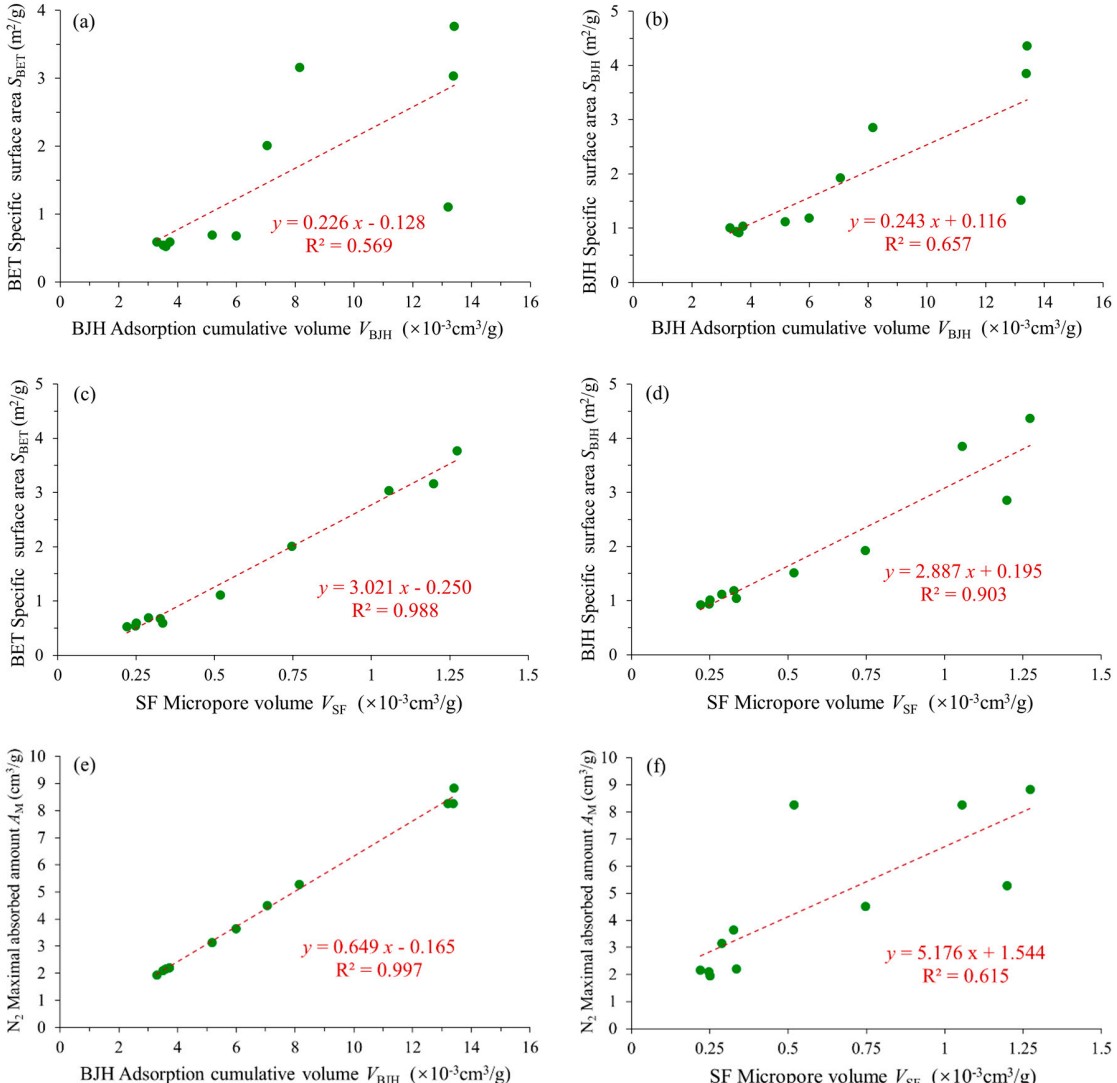

**Figure 6.** Univariate linear regression analysis of pore structure parameters: (**a**) Relationship between $V_{BJH}$ and $S_{BET}$; (**b**) relationship between $V_{BJH}$ and $S_{BJH}$; (**c**) relationship between $V_{SF}$ and $S_{BET}$; (**d**) relationship between $V_{SF}$ and $S_{BJH}$; (**e**) relationship between $V_{BJH}$ and $A_{M}$; (**f**) relationship between $V_{SF}$, and $A_{M}$.

A further discussion of the contribution rate ($V_r$) among five different size ranges pores to the total $S_{BJH}$ and total $V_{BJH}$ is conducted from the experimental result as showing in Figure 7. The average $V_r$ of micropore (<2 nm), type I mesopores (mesopore-I: 2–10 nm), type II mesopores (mesopore-II: 10–50 nm), type I macropores (macropore-I: 50–100 nm) and type II macropores (macropore-I: >100 nm) to the total $V_{BJH}$ are 0.48%, 23.21%, 37.14%, 17.63% respectively (Figure 7a). The average contribution rate of micropore, mesopore-I, mesopore-II, macropore-I and macropore-II to the total $S_{BJH}$ ($S_r$) is 3.05%, 72.65%, 20.08%, 3.21%, and 1.00%, respectively (Figure 7b). To summarize, BJH volume mainly hinges on the number of mesopore-II (37.14%), followed by the mesopore-II (23.21%) and macropore-II (21.54%). The pore type with the largest contribution rate for the total SSA is also mesopores, with an average value of 92.73% (mesopore-I: 76.25%, mesopore-II: 20.08%).

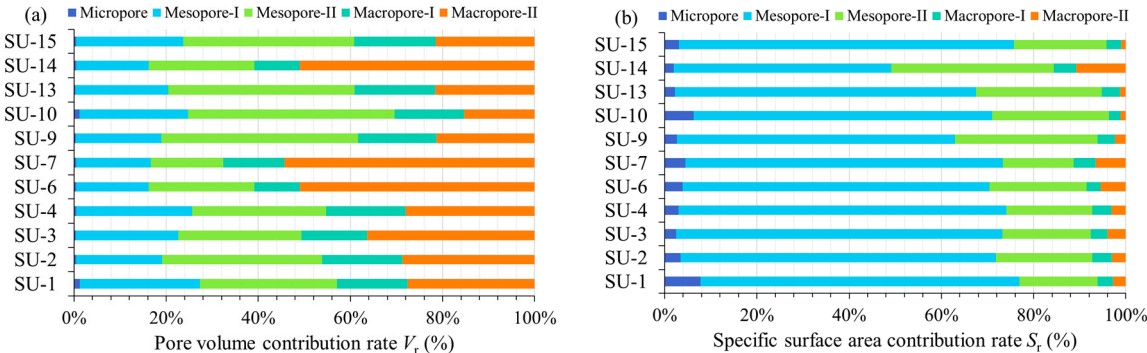

**Figure 7.** The contribution rate of $V_{BJH}$ and $S_{BJH}$ in five different size ranges pores. (**a**) Pore volume (PV) contribution rate of five different size classes pores; (**b**) specific surface area (SSA) contribution rate of five different size classes pores.

As illustrated in Figure 8, the plot of PSD of different Hysteresis loop types tight sandstone is showing a different characteristic. H1 type loops are dominated by mesopore-II (averages at 30.11%) and mesopore-I (averages at 23.05%), with one major pore size peak range of 2~50nm (Figure 8a1,b1). The H2 type loop is mainly controlled by the macropore-II (averages at 52.11%) and mesopore-II (averages at 20.57%), with two main pore size peak range of 10~50nm and 100~200nm (Figure 8a2,b2). In addition, the predominant pore size range of H3 type is mesopore –II (averages at 41.32%), which varies from 10–50nm (Figure 8a3,b3).

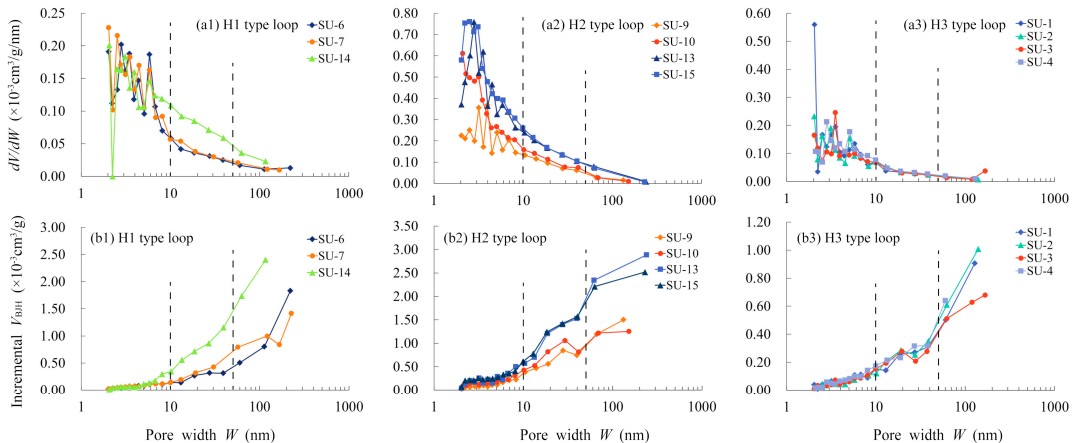

**Figure 8.** Representative pore size of different hysteresis loop types tight sandstone: (**a1**)–(**a3**) dV/dW *vs*. PW, (**b1**)–(**b3**) Incremental $V_{BJH}$ *vs*. PW.

### 3.4. The Fractal Characteristics of Nanoscale Pores

In this study, different piecewise linear fitting methods were used by selecting five different relative pressure ($P/P_0$) cutoff value, the ln($V$) vs. ln[ln($P_0/P$)] curves could be separated into two or three segments. The fractal fitting curves of three typical samples are illustrated in Figure 9, where the slope ($k$) was used to calculate fractal dimension value ($D_N$) according to Equation (1) and Equation (2), using calculated root mean square error ($R^2$) to show the fitting degree. As presented in Table 3, $D_N$ calculated by all four methods exhibit good fitting with $R^2$ values that are mostly higher than 0.97, and the calculated results are shown in Table 4.

In the first method, the unsegmented method was utilized to calculate $D_{N0}$, which can represent the heterogeneity of the whole nanoscale pores. The $D_{N0}$ values range from 2.360 to 2.592, averages at 2.446, which showing an ideal fractal feature in the nanoscale pore structure of He 8 Member tight sandstone.

Trichotomy was used in the second method. Three distinct fitting liner segments are obtained at different relative pressure corresponds to different BJH pore width ($W$) range respectively. The $D_{N1}$ ranges from 2.500 to 2.707 (averaging at 2.597), which is the value correspond to the pore microstructures complexity of pore size lager than 10 nm, while $D_{N2}$ is the representative value of pore structure heterogeneous features with PW range from 2 nm~10 nm, which is varied from 2.168 to 2.539 (averaging at 2.314). Correspondingly, $D_{N3}$ value represents the heterogeneity characteristic of micropores ($W < 2$ nm), ranges from 2.215 to 2.600 with an average of 2.382. The results show that the heterogeneity of macropores and micropores is generally stronger than that of mesopores.

The dichotomy was used to separate the $\ln(V)$ vs. $\ln[\ln(P_0/P)]$ plot into two segments with relative pressure point corresponds to the BJH $W = 5$ nm as the cutoff value, which can fit two straight line separately and obtained $D_{N4}$ (Pore size: $W > 5$ nm) and $D_{N5}$ (Pore size: $W \leq 5$ nm) severally. $D_{N4}$ values range from 2.481 to 2.660 with an average of 2.566. $D_{N5}$ values range from 2.063 to 2.191, averaging at 2.259, which is smaller than $D_{N4}$.

In the fourth method, selecting a cutoff value based on the inflection point of $N_2$-GA isotherms curve, divided into three the following areas: Region 1: $P/P_0 > 0.8$; Region 2: $0.5 < P/P_0 \leq 0.8$; Region 3: $P/P_0 \leq 0.5$, where $D_{N6}$ values ranges from 2.507 to 2.707, with an average of 2.602. $D_{N7}$ values range from 2.300 to 2.598, averaging at 2.417, which is lower than $D_{N6}$. $D_{N8}$ values range from 2.058 to 2.494, averaging at 2.261, which is smaller than $D_{N7}$.

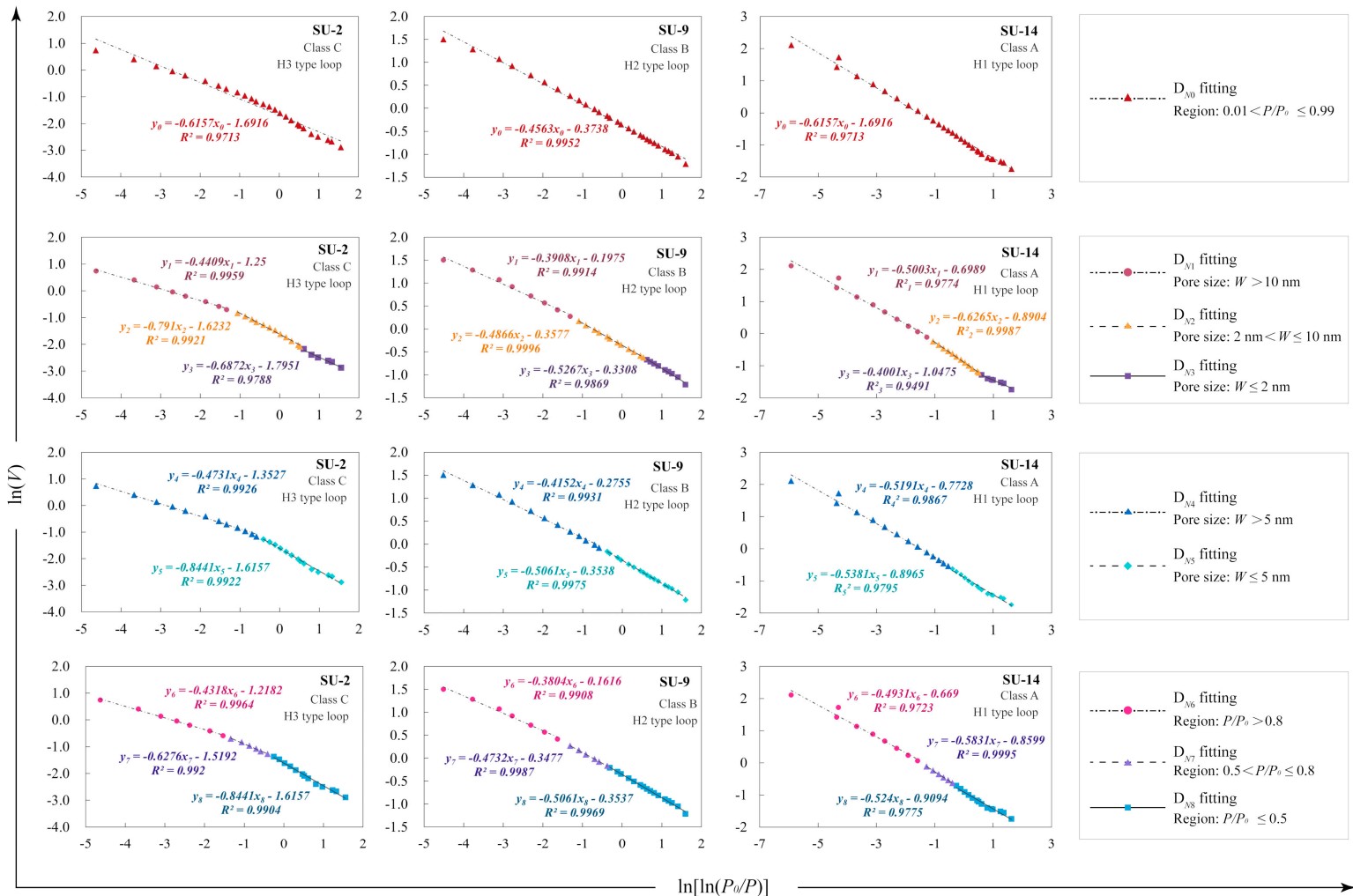

**Figure 9.** Representative plots of ln(*V*) vs. ln[ln(*P₀/P*)] reconstructed from the N₂ adsorption analysis of different class tight sandstone samples.

**Table 3.** Slop ($k$) and root mean square error ($R^2$) obtained from the plot of $\ln(V)$ *vs.* $\ln[(P_0/P)]$.

| Sample ID | Region: 0.01 < $P/P_0 \leq 0.99$ | | Pore size: $W > 10nm$ | | Pore size: 2 nm < $W \leq$ 10 nm | | Pore size: $W < 2nm$ | | Pore size: $W > 5nm$ | | Pore size: $W \leq 5nm$ | | Region 1: $P/P_0 > 0.8$ | | Region 2: 0.5 < $P/P_0 \leq 0.8$ | | Region 3: $P/P_0 \leq 0.5$ | |
|---|---|---|---|---|---|---|---|---|---|---|---|---|---|---|---|---|---|---|
| | $k_0$ | $R_0^2$ | $k_1$ | $R_1^2$ | $k_2$ | $R_2^2$ | $k_3$ | $R_3^2$ | $k_4$ | $R_4^2$ | $k_5$ | $R_5^2$ | $k_6$ | $R_6^2$ | $k_7$ | $R_7^2$ | $k_8$ | $R_8^2$ |
| SU-1 | −0.624 | 0.959 | −0.408 | 0.999 | −0.805 | 0.989 | −0.625 | 0.928 | −0.448 | 0.992 | −0.893 | 0.982 | −0.408 | 0.999 | −0.658 | 0.994 | −0.888 | 0.978 |
| SU-2 | −0.616 | 0.971 | −0.441 | 0.996 | −0.791 | 0.992 | −0.687 | 0.979 | −0.473 | 0.993 | −0.844 | 0.992 | −0.432 | 0.996 | −0.628 | 0.992 | −0.844 | 0.990 |
| SU-3 | −0.618 | 0.965 | −0.425 | 0.990 | −0.803 | 0.991 | −0.741 | 0.969 | −0.464 | 0.991 | −0.895 | 0.993 | −0.425 | 0.990 | −0.656 | 0.998 | −0.892 | 0.991 |
| SU-4 | −0.640 | 0.965 | −0.420 | 0.999 | −0.832 | 0.993 | −0.511 | 0.972 | −0.469 | 0.990 | −0.860 | 0.980 | −0.420 | 0.999 | −0.700 | 0.996 | −0.851 | 0.975 |
| SU-6 | −0.591 | 0.962 | −0.431 | 0.994 | −0.807 | 0.994 | −0.762 | 0.971 | −0.445 | 0.995 | −0.895 | 0.994 | −0.435 | 0.993 | −0.689 | 0.997 | −0.897 | 0.993 |
| SU-8 | −0.614 | 0.965 | −0.467 | 0.985 | −0.793 | 0.990 | −0.786 | 0.962 | −0.472 | 0.993 | −0.937 | 0.991 | −0.467 | 0.985 | −0.643 | 0.996 | −0.942 | 0.989 |
| SU-9 | −0.456 | 0.995 | −0.391 | 0.991 | −0.487 | 1.000 | −0.527 | 1.000 | −0.415 | 0.993 | −0.506 | 0.998 | −0.380 | 0.991 | −0.473 | 0.999 | −0.506 | 0.997 |
| SU-10 | −0.408 | 0.982 | −0.316 | 0.978 | −0.461 | 0.995 | −0.525 | 0.973 | −0.340 | 0.986 | −0.525 | 0.996 | −0.305 | 0.974 | −0.402 | 1.000 | −0.528 | 0.996 |
| SU-13 | −0.483 | 0.972 | −0.335 | 0.985 | −0.602 | 0.993 | −0.566 | 0.992 | −0.379 | 0.958 | −0.611 | 0.997 | −0.318 | 0.929 | −0.505 | 0.999 | −0.610 | 0.996 |
| SU-14 | −0.548 | 0.994 | −0.500 | 0.977 | −0.627 | 0.999 | −0.400 | 0.949 | −0.519 | 0.987 | −0.538 | 0.980 | −0.493 | 0.972 | −0.583 | 1.000 | −0.524 | 0.978 |
| SU-15 | −0.498 | 0.963 | −0.293 | 0.931 | −0.542 | 0.994 | −0.669 | 0.970 | −0.347 | 0.959 | −0.651 | 0.993 | −0.293 | 0.931 | −0.471 | 1.000 | −0.651 | 0.993 |
| Minimum | −0.640 | 0.959 | −0.500 | 0.931 | −0.832 | 0.989 | −0.786 | 0.928 | −0.519 | 0.958 | −0.937 | 0.980 | −0.493 | 0.929 | −0.700 | 0.992 | −0.942 | 0.975 |
| Maximum | −0.408 | 0.995 | −0.293 | 0.999 | −0.461 | 1.000 | −0.400 | 1.000 | −0.340 | 0.995 | −0.506 | 0.998 | −0.293 | 0.999 | −0.402 | 1.000 | −0.506 | 0.997 |
| Average | −0.554 | 0.972 | −0.403 | 0.984 | −0.686 | 0.994 | −0.618 | 0.970 | −0.434 | 0.985 | −0.741 | 0.990 | −0.398 | 0.978 | −0.583 | 0.997 | −0.739 | 0.989 |

**Table 4.** Fractal dimensions from $N_2$ adsorption data of different class He 8 Member tight sandstone samples.

| Sample ID | Hysteresis Loop Types | $D_{N0}$ | $D_{N1}$ | $D_{N2}$ | $D_{N3}$ | $D_{N4}$ | $D_{N5}$ | $D_{N6}$ | $D_{N7}$ | $D_{N8}$ |
|---|---|---|---|---|---|---|---|---|---|---|
| SU-1 | H3 | 2.376 | 2.592 | 2.195 | 2.375 | 2.552 | 2.107 | 2.592 | 2.342 | 2.112 |
| SU-2 | H3 | 2.384 | 2.559 | 2.209 | 2.313 | 2.527 | 2.156 | 2.568 | 2.372 | 2.156 |
| SU-3 | H3 | 2.382 | 2.575 | 2.197 | 2.259 | 2.536 | 2.105 | 2.575 | 2.344 | 2.108 |
| SU-4 | H3 | 2.360 | 2.580 | 2.168 | 2.490 | 2.531 | 2.140 | 2.580 | 2.300 | 2.150 |
| SU-6 | H1 | 2.409 | 2.569 | 2.193 | 2.238 | 2.555 | 2.105 | 2.565 | 2.311 | 2.103 |
| SU-7 | H1 | 2.386 | 2.533 | 2.207 | 2.215 | 2.528 | 2.063 | 2.533 | 2.357 | 2.058 |
| SU-14 | H1 | 2.452 | 2.500 | 2.374 | 2.600 | 2.481 | 2.462 | 2.507 | 2.417 | 2.476 |
| SU-9 | H2 | 2.544 | 2.609 | 2.513 | 2.473 | 2.585 | 2.494 | 2.620 | 2.527 | 2.494 |
| SU-10 | H2 | 2.592 | 2.685 | 2.539 | 2.475 | 2.660 | 2.475 | 2.695 | 2.598 | 2.472 |
| SU-13 | H2 | 2.517 | 2.665 | 2.398 | 2.434 | 2.621 | 2.389 | 2.682 | 2.496 | 2.390 |
| SU-15 | H2 | 2.502 | 2.707 | 2.458 | 2.331 | 2.653 | 2.349 | 2.707 | 2.529 | 2.349 |
| Minimum | | 2.360 | 2.500 | 2.168 | 2.215 | 2.481 | 2.063 | 2.507 | 2.300 | 2.058 |
| Maximum | | 2.592 | 2.707 | 2.539 | 2.600 | 2.660 | 2.494 | 2.707 | 2.598 | 2.494 |
| Average | | 2.446 | 2.597 | 2.314 | 2.382 | 2.566 | 2.259 | 2.602 | 2.417 | 2.261 |

## 4. Discussion

### 4.1. Effect of Mineral Composition on Fractal Dimensions

The heterogeneous distribution of mineral particles constitutes a diversiform pore-throat system. FE-SEM images show that pore types of all tight sandstone samples consist of different mineral edge can be classified into four types: original intergranular pore (Figure 10a), secondary dissolved intragranular pore (Figure 10b), intercrystalline micropores of clay minerals (Figure 10c–f), and microfissure (Figure 10a).

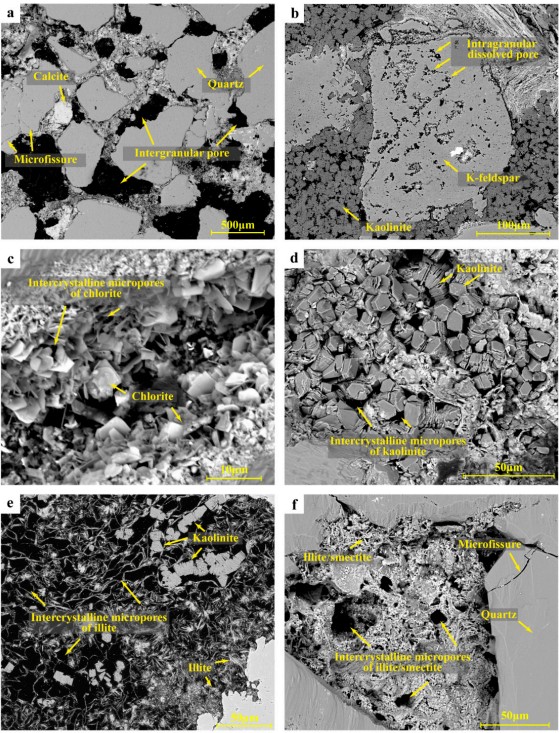

**Figure 10.** Field emission scanning electron microscopy (FE-SEM) images of different pore types. (**a**) Intergranular pores; (**b**) intragranular pores; (**c**) intercrystalline micropores of chlorite; (**d**) intercrystalline micropores of kaolinite; (**e**) intercrystalline micropores of illite; (**f**) intercrystalline micropores of illite/smectite.

Quartz is the highest mineral composition in all tight sandstone samples according to the X-RD results (Table 1 and Figure 3) and FE-SEM images (Figure 10a). Therefore, it is extremely important to point out the relationship between quartz and $D_N$. The result shows that quartz content is showing a good negative correlation with all types of fractal dimensions respectively, which $D_{N0}$ and $D_{N2}$ showing the best correlation with quartz content (Figure 11a,b), with correlation coefficients of 0.751 and 0.748. A higher quartz content indicates the pore-throat system tend to be simpler, which is contrary to the relationship between total clay mineral content and $D_N$ (Figure 11c,d), with a positive correlation coefficient of 0.865 ($D_{N0}$) and 0.809 ($D_{N7}$). This is associated with the extensive distribution of clay minerals and its complex composition, which provides a large amount of nano-scale storage space for the occurrence of tight gas and makes the pore-throat system more complex at the same time.

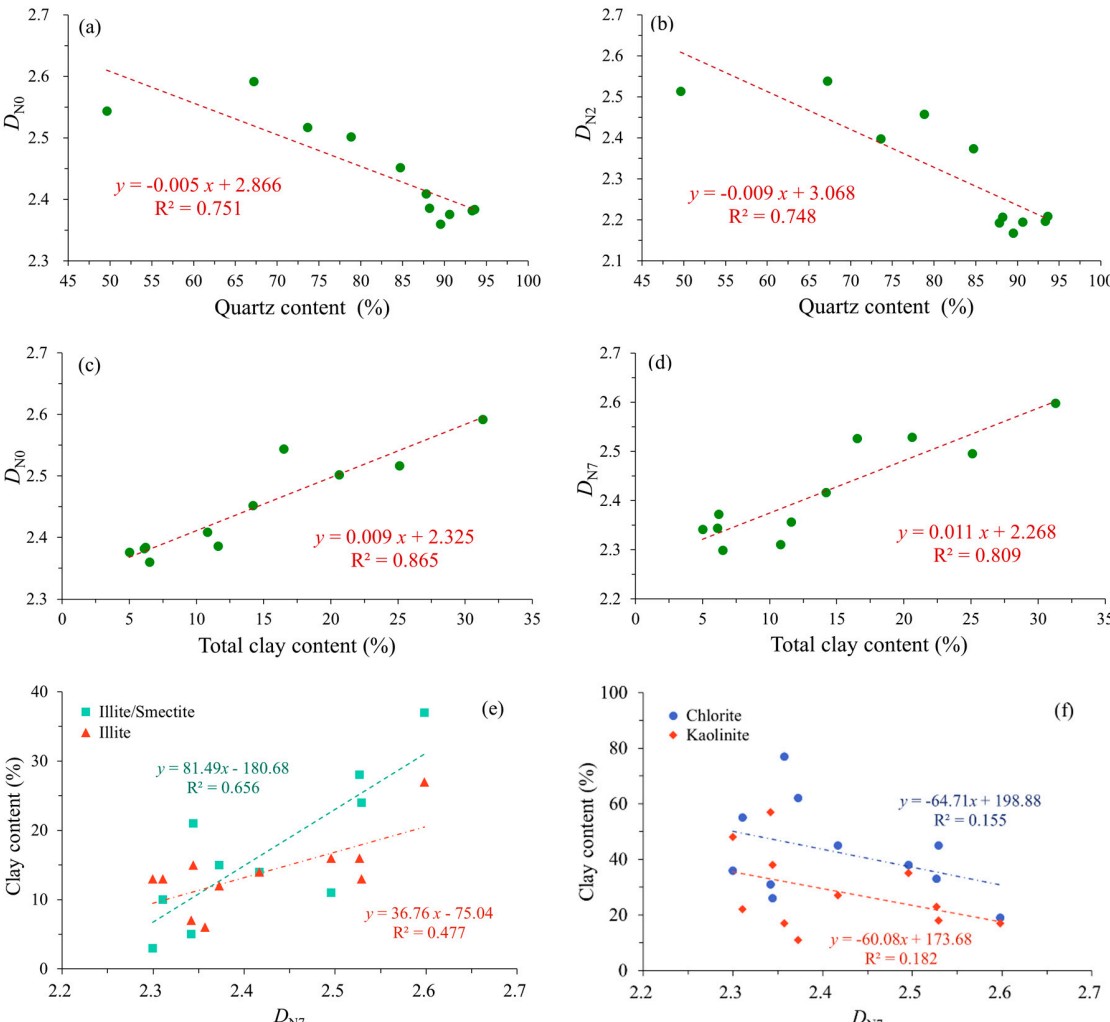

**Figure 11.** The effect of mineral content on fractal dimensions. (**a**) The relationship between quartz content and $D_{N0}$; (**b**) the relationship between quartz content and $D_{N2}$; (**c**) the relationship between total clay content and $D_{N0}$; (**d**) the relationship between total clay content and $D_{N7}$; (**e**) the relationship between illite/smectite and illite with $D_{N7}$; (**f**) the relationship between chlorite and kaolinite with $D_{N7}$.

In addition, the intercrystalline micropores of chlorite and kaolinite has a smaller SSA than that is in illite/smectite and illite because it has a smoother edge. It also makes the illite and illite/smectite has good positive correlation with fractal dimensions $D_{N7}$ (Figure 11e,f), where the phenomenon of mixed layer mineral is more obvious.

### 4.2. Effect of Pore Size Distribution on Fractal Dimensions

Pore-throat system is a crucial storage space of tight sandstone gas as well as significant flowing channel for natural gas during the entire development process. This means that mineral composition has a significant influence on SSA, PV, and $D_N$, respectively. As shown in Figure 12a, all the tight sandstone samples are arranged in the order of the clay mineral content increasing; the total $S_{BET}$ also has an increasing trend identified. Furthermore, $S_{BET}$ has an excellent positive relationship with all types of fractal dimensions (P-value < 0.01, except of $D_{N5}$ and $D_{N8}$), where the correlation with $D_{N7}$ ($R^2 = 0.783$), $D_{N4}$ ($R^2 = 0.794$) and $D_{N2}$ ($R^2 = 0.783$) is more obvious. Similarly, as the increasing of clay mineral content, PV also has an increasing tendency (Figure 12b). Moreover, the relationship between $D_N$ and SF micropores volume is significant (P-value < 0.01, except of $D_{N5}$, $D_{N8}$, and $D_{N3}$), with a positive correlation coefficient of 0.844 ($D_{N7}$) and 0.794 ($D_{N0}$), whereas it is not significant between the relationship of $D_N$ and $V_{BJH}$, with a positive correlation coefficient of 0.416 ($D_{N7}$) and 0.406 ($D_{N0}$).

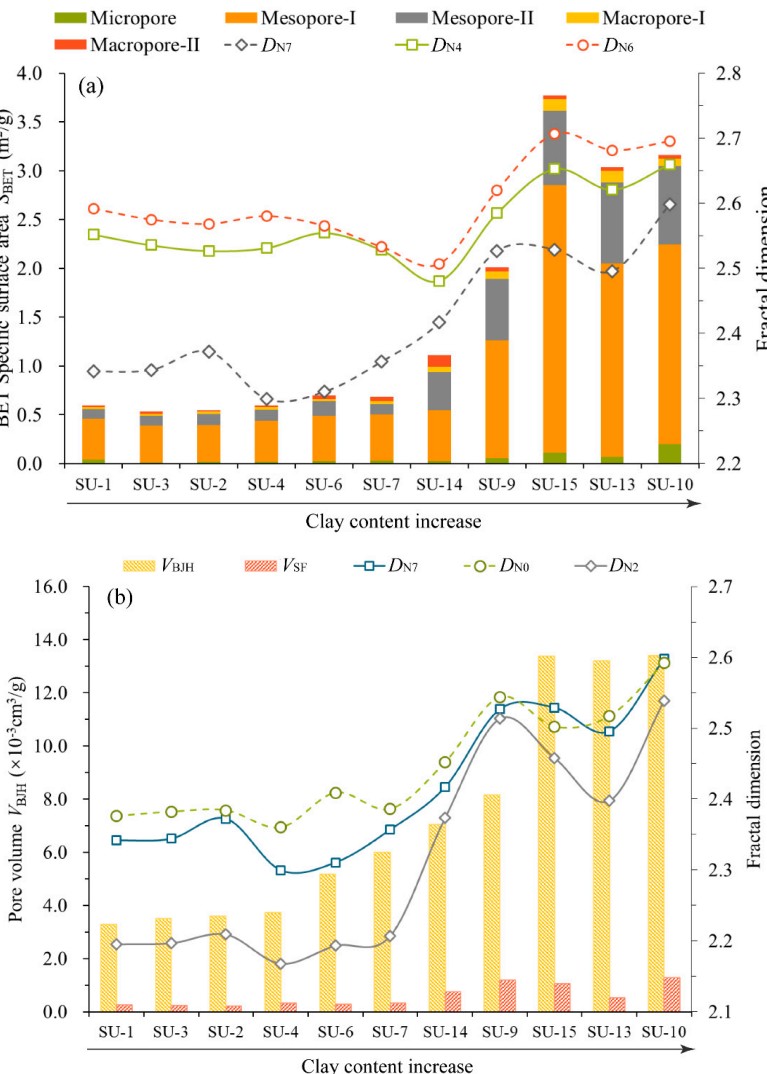

**Figure 12.** The relationship between pore structure parameter and Fractal dimensions in the order of clay mineral content increasing. (**a**) The relationship between $S_{BET}$ and fractal dimensions; (**b**) the relationship between PV and fractal dimensions.

Moreover, the relationship between the SSA contribution rate ($S_r$) of different pore size range with fractal dimensions shows the different characters. Thereinto, the mesopore-II has an excellent positive relationship with $D_{N8}$ ($R^2 = 0.746$, P-vuale < 0.01) (Figure 13a), whereas macropore-II has a negative

correlation with $D_{N1}$ ($R^2 = 0.730$,P-vuale < 0.01) (Figure 13b), this result shows that where there is a smaller $S_r$ of mesopore-II and a lager $S_r$ of macropore-II, there is a simpler pore-throat structure, such as SU-14.

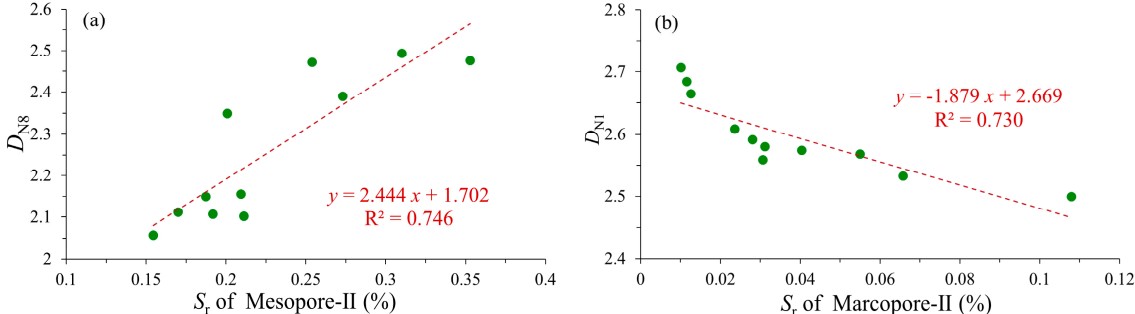

**Figure 13.** The relationship between $S_r$ and $D_N$. (**a**) The relationship between $Sr$ of mesopore-II and $D_{N8}$; (**b**) the relationship between $Sr$ of macropore-II and $D_{N1}$.

### 4.3. Competive Study of Fractal Dimensions

As illustrated in Figures 9 and 14, the unsegmented and segmented linear fitting methods both are effective way to obtain the fractal curves based on $N_2$-GA isotherms curve, which can accurately represent the fractal characteristic of the different pore size range separately and indicate that the distribution of the pore throat in He 8 Member tight sandstone has multi-fractal characteristic. All types of fractal dimensions showing an increasing trend as the clay mineral content increasing, which indicate a good relationship between mineral composition with $D_n$. Thereinto, the value of $D_{N0}$ is the closest to the average and the $D_{N2}$ has the best relationship with the average value ($R^2 = 0.954$).

Furthermore, as listed in Table 4, the fractal dimensions always appear the highest average value in tight sandstone sample with H2 type hysteresis loops no matter what methods were used, indicating that pore structure heterogeneity is mainly controlled by the inkbottle-shaped pores and the $V_r$ of mesopore.

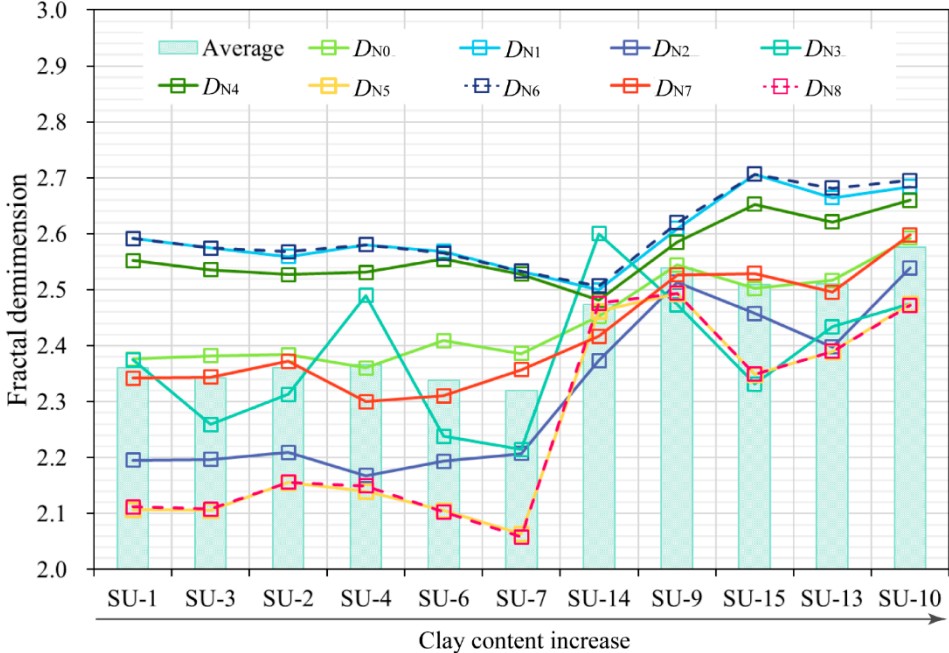

**Figure 14.** The trends of fractal dimension among different tight sandstone samples.

Among different types of fractal dimensions, the overall distribution of $D_{N0}$, $D_{N1}$, $D_{N4}$, and $D_{N6}$ is more concentrated, whereas the whole distribution of $D_{N2}$, $D_{N3}$, $D_{N5}$, $D_{N7}$ and $D_{N8}$ is more scattered (Figure 15). This is relevant to the heterogeneity of pore structure and mineral composition, which expanded the range of fractal dimension values, in other words, this type of fractal dimensions is more sensitive to certain specific property. Correlation analysis results show that expect of $D_{N3}$ showing a weak relationship with chlorite content ($R^2$ = 0.252), which present the best correlation among all types of $D_N$, all the "scattered" fractal dimensions is bound up with mineral content to some extent, while "concentrated" fractal dimensions is much weaker except of $D_{N0}$. In addition, $D_{N5}$ and $D_{N8}$ merely present a good relationship with mineral content (especially in quartz content and illite/smectite content) (Figures 11 and 14) and $S_r$ of mesopore-II, which is optimum parameter to represent the complexity of mineral composition and its corresponding micropore shape.

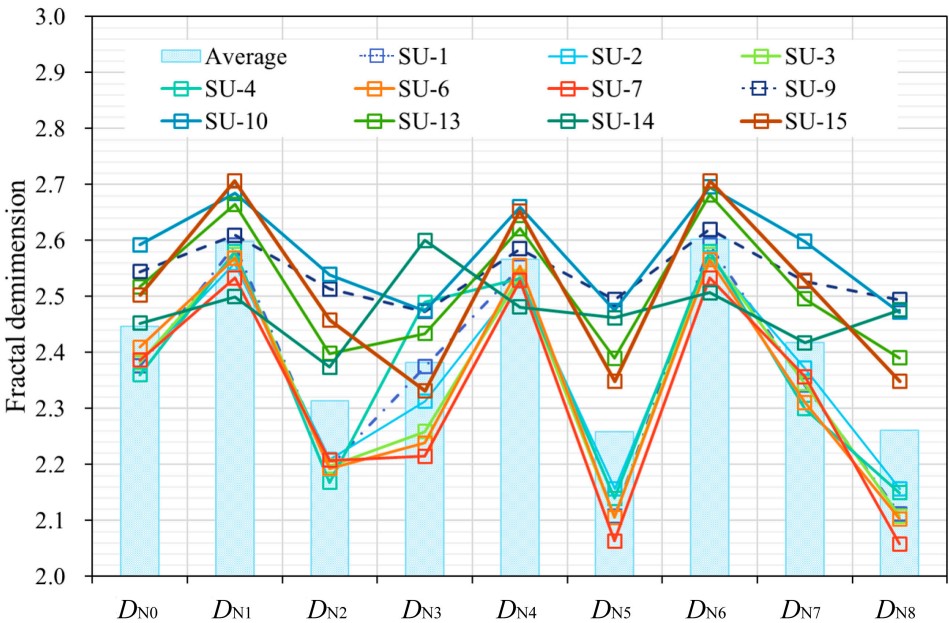

**Figure 15.** Different types of fractal dimension numerical distribution range in different tight sandstone.

## 5. Conclusions

In this paper, nine types fractal dimension were obtained using different piecewise linear fitting methods by selecting five different relative pressure ($P/P_0$), and we put forward the concept of "concentrated" fractal dimensions and "scattered" fractal dimensions for the first time according to its concentration extent of distribute in different samples. Moreover, three types of hysteresis loops were distinguished that correspond to different pore shape combination and the impact of mineral composition and pore structure on different fractal dimensions were discussed in detail. These findings can provide new insights for quantitative evaluation the microscopic heterogeneity of tight sandstone reservoir at various pore size classes and further reveal the controlling effect of pore heterogeneity on natural gas storage and desorption, which is of great significance for improving the exploitation efficiency of tight sandstone gas. The main conclusions are summarized as follows:

(1) Among all types of fractal dimensions, $D_{N0}$ represent the heterogeneity of whole nanoscale pore, with the ranges of 2.360 to 2.592, averaging at 2.446, which show an ideal fractal feature of the nanoscale pore structure of tight sandstone in He 8 Member. Meanwhile, the value of $D_{N0}$ is the closest to the average while the $D_{N2}$ has the best relationship with the average value ($R^2$ = 0.954).

(2) The "scattered" fractal dimensions ($D_{N2}$, $D_{N3}$, $D_{N5}$, $D_{N7}$) are more sensitive to certain specific property of the reservoir, including mineral content (especially in quartz content and illite/smectite content) and the $S_r$ of mesopore-II (10–50 nm), where $D_{N5}$ and $D_{N8}$ are optimum parameter to represent the complexity of mineral composition and its corresponding micropore shape.

(3) H1 type loop indicated the combination of cylindrical pores and parallel plate shaped pores; H2 type loop related to the combination of inkbottle-shaped pores and cylindrical pores, with bowl-shaped pores as a transitional form in the mesopores; H3 type loop is associated with the combination of slit-/wedge-shaped pores and parallel plate shaped pores. Furthermore, the heterogeneity of pore structure in this study area is predominantly controlled by inkbottle-shaped pores and the $V_r$ of mesopores.

To sum it up, the analysis of fractal dimensions is a topic of actuality in geosciences in general and contributes to a better understanding of methane adsorption characteristics in tight sandstone gas resources. This work addressed the characteristics of volume fractal dimensions of tight sandstone at various size classes, but the surface fractal dimensions are not discussed in this paper, which could be a future research direction.

**Author Contributions:** Data curation, Z.W. and X.J.; Formal analysis, Z.W.; Investigation, Z.W.; Methodology, M.P. and Y.S.; Project administration, Y.S.; Resources, Y.S.; Supervision, M.P. and Y.S.; Visualization, Z.W.; Writing—original draft, Z.W.; Writing—review & editing, X.J. All authors have read and agreed to the published version of the manuscript.

**Funding:** This research was funded by the National Science and Technology Major Project of China, grant number 2017ZX05013005-009.

**Acknowledgments:** The authors greatly appreciate the Changqing Oilfield Branch Company of China National Petroleum Corporation for all the support during this work. The authors would like to thank all editors and reviewers for their constructive comments on this paper.

**Conflicts of Interest:** The authors declare no conflict of interest.

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
