# Peer review of "Nano-Scale Pore Structure and Its Multi-Fractal Characteristics of Tight Sandstone by N2 Adsorption/Desorption Analyses: A Case Study of Shihezi Formation from the Sulige Gas Filed, Ordos Basin, China"

_minerals, doi:10.3390/min10040377_

Round 1
Reviewer 1 Report
This manuscript describes a novel application of the fractal dimension method to characterize the pore structure of tight sandstones, which is relevant in regards to the study of natural gas resources. The manuscript describes N2 sorption experiments, X-ray diffraction experiments and field emission-scanning electron microscopy observations. Results indicate that the pore structure is dominated by the contribution from the mesopores range.
My main impression is that the article is interesting and has sufficient impact to add to the knowledge base. Here below are some specific comments:
- The article conforms to the journal instructions and scope.
- The analysis of fractal dimensions is a topic of actuality in the Geosciences in general. In particular, it helps to gain a better understanding about the pore structure of tight rocks.
- However, I have the sense that another round of revision by an English-native professional editor could help to improve the readability of the manuscript. For example, I wonder if the usage of connectors and punctuation signs are employed correctly in the following lines: 55, 63, 77, 83, 94, 118, and 122.
I think that the abstract needs a couple of edits to remark more strongly certain key elements, such as research limitations, practical implications, and originality or value of the work. My impression is that the abstract doesn’t reveal well the content of the manuscript.
Author Response
Response to Reviewer 1 Comments
Thanks for your valuable comments and suggestions for our manuscript entitled "Nano-scale pore structure and its Multi-fractal Characteristics of Tight Sandstone by N2 Adsorption/Desorption Analyses: A Case Study of Shihezi Formation from the Sulige Gas Filed, Ordos Basin, China" (ID: minerals-760684). It is very helpful for revising and improving the presentation effect of our study result. We have revised our manuscript using the "Track Changes" function in Microsoft Word after a careful consideration of your suggestions, follows are the responds to your comments point by point:
Point 1: I have the sense that another round of revision by an English-native professional editor could help to improve the readability of the manuscript. For example, I wonder if the usage of connectors and punctuation signs are employed correctly in the following lines: 55, 63, 77, 83, 94, 118, and 122.
Response 1: We have checked the grammar and spelling of the full manuscript comprehensively according to your suggestion and made some corresponding corrections. The usage of abbreviation: ‘N2-GA’ in our manuscript is adhering to the abbreviation of reference [18] and [28].
Point 2: I think that the abstract needs a couple of edits to remark more strongly certain key elements, such as research limitations, practical implications, and originality or value of the work. My impression is that the abstract doesn’t reveal well the content of the manuscript.
Response 2: We have recognized that the abstract are not comprehensive so that we’ve highlight the value and the implications of our research under your suggestion. In addition, we’ve added the practical imitation of our research in the conclusion part. In addition, We have added a Literature Review section in the introduction part to help readers knowing the background of our research subject.
Reviewer 2 Report
The paper demonstrated the role played by the nanopore structure of sandstone in methane adsorption with different experimental and numerical methods.
The topic is of practical interest in the oil and gas industry and the theory behind the results is sound and well explained.
Overall, the paper has been fairly organized and presented. However, the paper requires some changes before it can be published.
A Literature Review section should be developed to motivate the readers in the background of the research subject.
Among others the reviewer suggest to take into consideration the following paper: Giacchetta G., Leporini , Marchetti B., 2015, Economic and environmental analysis of a Steam Assisted Gravity Drainage (SAGD) facility for oil recovery from Canadian oil sands, Applied Energy, Vol. 142: 1–9
The Conclusions should be revised in order to highlight the unique contributions of the paper, limitations of the research and some future research directions.
Author Response
Response to Reviewer 2 Comments
Thanks for your valuable comments and suggestions for our manuscript entitled “Nano-scale pore structure and its Multi-fractal Characteristics of Tight Sandstone by N2 Adsorption/Desorption Analyses: A Case Study of Shihezi Formation from the Sulige Gas Filed, Ordos Basin, China” (ID: minerals-760684). It is very helpful for revising and improving the presentation effect of our study result. We have revised our manuscript using the “Track Changes” function in Microsoft Word after a careful consideration of your suggestions, follows are the responds to your comments point by point:
Point 1: A Literature Review section should be developed to motivate the readers in the background of the research subject.
Response 1: We have added a Literature Review section in the introduction part to help readers knowing the background of our research subject.
Point 2: The reviewer suggest to take into consideration the following paper: Giacchetta G., Leporini , Marchetti B., 2015, Economic and environmental analysis of a Steam Assisted Gravity Drainage (SAGD) facility for oil recovery from Canadian oil sands, Applied Energy, Vol. 142: 1–9
Response 2: We have read this paper carefully; this work addressed the role of the economic and environmental feasibility on the Canadian Oil Sands exploration and recovery. We have referred to the writing methods and learned the main conclusions of this paper, cited it in the introduction part (line 37).
Point 3: The Conclusions should be revised in order to highlight the unique contributions of the paper, limitations of the research and some future research directions.
Response 3: We have recognized that the conclusions are not comprehensive so that we have highlight the unique contributions of our paper under your suggestion. We also illustrated limitations of of the research and proposed a future research direction at line 385 as well.